# Mobility-Aware Proactive Edge Caching Optimization Scheme in Information-Centric IoV Networks

**DOI:** 10.3390/s22041387

**Published:** 2022-02-11

**Authors:** Salahadin Seid Musa, Marco Zennaro, Mulugeta Libsie, Ermanno Pietrosemoli

**Affiliations:** 1Department of Computer Science, Addis Ababa University, Addis Ababa P.O. Box 1176, Ethiopia; salubinseid@gmail.com (S.S.M.); mulugeta.libsie@aau.edu.et (M.L.); 2STI Unit, Abdus Salam International Centre for Theoretical Physics, 34151 Trieste, Italy; ermanno@ictp.it

**Keywords:** internet of vehicles, proactive content caching, mobility, vehicular caching, roadside unit, edge caching, information centric networks, QoS

## Abstract

Edge caching is a promising approach to alleviate the burden on the backhaul of network links. It has a significant role in the Internet of Vehicle (IoV) networks performance by providing cached data at the edge and reduce the burden of the core network caused by the number of participating vehicles and data volume. However, due to the limited computing and storage capabilities of edge devices, it is hard to guarantee that all contents are cached and every requirement of the device are satisfied for all users. In this paper, we design an Information-Centric Network (ICN) with mobility-aware proactive caching scheme to provide delay-sensitive services on IoV networks. The real-time status and interaction of vehicles with other vehicles and Roadside Units (RSU) is modeled using a Markov process. Mobility aware proactive edge caching decision that maximize network performance while minimizing transmission delay is applied. Our numerical simulation results show that the proposed scheme outperforms related caching schemes in terms of latency by 20–25% in terms of latency and by 15–23% in cache hits.

## 1. Introduction

With the advancement of wireless communication, edge computing and caching, as well as the Internet of Things, the automotive industry continues to accelerate towards the goal of connected vehicles, named Internet of Vehicles (IoV). IoV is an active research area and has been considered a key enabling technology to provide the Intelligent Transport Systems (ITS) required to improve road traffic efficiency, enhance road safety, and reduce traffic congestion. Moreover, IoV networks promise to enable a wide range of applications including infotainment and road safety applications [1]. To improve the driving and traveling experience, significant volumes of information will be exchanged between vehicles and roadside units in the IoV ecosystem. Consequently, these applications require specific computational and communication resources such as bandwidth and storage, while keeping low latency to meet the Quality of Experience (QoE) of IoV users. In IoV networks the movement of vehicles causes dynamic topology changes which require the rerouting of the content. These dynamic topology changes due to the mobility of vehicles, added to the poor quality of the wireless links, are the hurdles to overcome in order to provide optimal IoV services.

ICN has been proposed to address the shortcomings of the current host-centric Internet architecture. It is an emerging network paradigm that decouples content from its storage location by distributing one or more copies of the content across the network [2]. The dynamic nature of IoV and factors such as device mobility and possible link failure, changing network bandwidth requirements, and dynamic data exchange, demand a new Internet architecture beyond the current TCP/IP based networks. ICN makes the content directly addressable and routable in the network. Enabling ICN with edge computing can improve the efficiency of content distribution and communication performance by reducing the distance between users and services [3].

Caching techniques have been used in nearly every aspect of communication networks. Recently, edge computing and caching have been introduced as technologies to provide low latency services alleviating loads and reducing content transmission delay by migrating cloud services to the network edge. Caching popular content at the wireless edge, including vehicles, Road Side Units (RSUs), and Base Stations (BSs), is an effective approach to alleviate the heavy burden on backhaul links, as well as to achieve lower delays and deployment costs. Thus, it promises to increase the performance of applications accessing cached data at the edge and relieve the core network by addressing the increasing bandwidth demands caused by large data volume and the number of participating vehicles. Caching is expected to play a significant role in storage-assisted Internet architectures, ICNs, and wireless systems in the near future, lowering operational and capital costs while boosting user services.

However, due to the limited computing and storage capabilities of edge devices, it is hard to guarantee that all contents are cached and every requirement of the device are satisfied for all users. Recently, modeling of efficient caching scheme strategies has received much attention from researchers [4,5]. Reactive caching [6] and proactive caching [7] are the common approaches to cache the contents. However, to leverage the advantage of proactive caching, a strategy for optimal caching decisions at the RSU is needed.

Addressing the above problems, we propose mobility-aware proactive caching in ICN for IoV. The real-time status of vehicles and their interaction with other vehicles and RSUs is modeled using a Markov process [8,9]. We propose an in-network edge caching that maximizes network performance while minimizing transmission delay. This is formulated as an optimization problem, and comparisons with existing related works are presented. Results show that our proposed scheme achieves high cache hit ratios and lower delays compared to conventional caching schemes in the context of high mobility edge caching IoV networks. The following are the main contributions of this paper:We model the ICN based IoV networks with mobility aware proactive caching scheme where both vehicles and RSUs have caching capabilities;We model the real-time status and mobility of nodes using a Markov process;We formulate an optimal caching problem to minimize the system cost, addressing limited storage capabilities of both RSUs and vehicles, while considering the constraints of vehicles mobility and latency requirements. Then we develop an algorithm to configure the caching placement as well as to determine the sets of possible RSUs and neighboring vehicles;We present simulation results to illustrate the performance of the algorithm. We used the SUMO mobility simulator to model vehicle mobility. We evaluated our scheme using latency and cache hit ratio metrics and compared it with other related caching approaches. Experimental results reveal that in high mobility scenarios our caching schemes show a significant performance gain in reducing total delay.

The rest of the paper is organized as follows. Section 2 briefly describes related works in mobility-aware caching. Section 3 describes the system model and problem formulation. The proposed caching algorithm is discussed in Section 4. Numerical simulation results are presented in Section 5. Section 6 presents conclusions and provides suggestions for future research.

## 2. Background and Related Works

### 2.1. ICN Basic and Caching

Edge caching has been studied extensively in response to the increasing demand for efficient content distribution. In this Section, we briefly review edge caching works from the perspective of ICN [2], IoV communication, and networks.

Caching is the process of storing data in a local memory, a well known technique that has been widely applied in Web access and more recently in Content Delivery Networks (CDN). The main objective is to improve QoE and reduce network and/or backhaul server congestion, decreasing delay and load. In CDN popular content is replicated on many servers spread throughout the Internet to reduce latency and traffic load [10]. However, CDN also has its own drawbacks and limitations, in terms of manageability, scalability, mobility, and security.

Named Data Networking (NDN) is an alternative to existing host-centric IP architecture and is a perfect candidate for content-centric edge computing. Differing from IP networks that use IP addresses to identify where contents are located, the basic idea of NDN is to retrieve the named pieces of information from any node that has it cached and provide it to the user. According to the recent NDN developer guide [11], each NDN node can store content, manifestations of incoming interest from the consumer, and forwarding information using three data structures: Content Store (CS), Pending Interest Table (PIT), and Forwarding Interest Table (FIB). A node receiving an interest for content performs a CS lookup on the content name. If the content is available in its cache, it returns the content and drops the interest. However, if the content is not available in the CS, the node performs a lookup in its PIT to check whether there is an existing entry for it. If the PIT lookup is successful, the router adds the incoming interest’s interface to the PIT entry (interest aggregation) and drops the interest. If no PIT match is found, the router creates a new PIT entry for the interest and forwards the interest using the FIB to an upstream router in the direction of the data source(s). Content Store (CS) is a cache of data packets. It provides methods to insert a data packet, find cached data that matches an Interest, and enumerate cached Data. NDN Forwarding Daemon (NFD) offers multiple cache replacement policies including a priority policy and an LRU (Least Recently Used) policy, which can be selected in the NFD configuration file.

Figure 1 illustrates the use of caching in ICN based IoV networks. The consumer vehicle initiates the communication to get the required content following a pull-based paradigm. It will generate an interest packet and broadcast this packet to the potential nodes that meet its requirement. Any intermediate node (another vehicle or an RSU) that caches the content will send a data packet to the consumer vehicle. At the beginning, consumer vehicle A broadcasts its interest packet and vehicle B forwards the interest packet to an RSU, then to macro base station and finally to the producer. The producer seeks out the matching information and returns the encapsulated data packet down to the consumer vehicle. Upon receiving the backward data packet, nodes between the producer and consumer send the encapsulated content back to the consumer vehicle while caching the content unit to its local cache for future use. When another consumer vehicle request the same content and sends the interest packet, intermediate node can detect the interest and satisfy the demand of the new consumer vehicle. Thus, the interest packet of the new consumer vehicle is not sent to the producer since its demand is satisfied by any one of the in-network nodes. It can be noted that caching in ICN for IoV networks decrease the number of redundant interest and data packets and improves transmission efficiency. The inherent in-network caching function of ICN promises to revolutionize the future network architecture. Thus, it has the potential to significantly reduce the transmission delay and traffic load in the next generation network [12].

### 2.2. Related Works

In this subsection, we present related works on edge caching in general and mobility-aware proactive caching scheme in particular.

Information demand has risen considerably in recent years as a result of the growth of mobile data, and network traffic has increased tremendously. As a result, communication delay becomes an important factor to consider. Edge caching is a potential solution that allows for the most frequently accessed and popular content to be cached on edge nodes, reducing backhaul network congestion. However, because edge nodes have limited cache storage, a content caching technique is required and should be optimized to reduce cache redundancy. Especially in high mobility IoV scenarios, it is difficult to decide which content needs to be cached at which location to attain effective and efficient results. A number of researchers use the mobility pattern of vehicles for prediction and selection of the edge node to cache contents [13,14]. Markov models have been used to predict the next location based on the mobility behavior of the mobile user. The work in [15] used three mobility datasets to demonstrate the efficiency of incorporating the previous *n* visited places of the user to increase the prediction accuracy of the next place. Mobility-aware proactive content caching for moving vehicles has been introduced in [13]. The main focus of that work is to reduce communication latency and improve QoS for vehicular networks in content-centric vehicular networks.

The work in [3] makes a performance comparison of caching strategies for information-centric IoT. The work in [14] present a proactive caching strategy that leverages ICN’s flexibility of caching data anywhere in the network, rather than just at the edge, as in conventional CDN networks. The paper uses entropy to measure mobility prediction uncertainty and locate the best prefetching node, thus eliminating redundancy. While prefetching at levels higher in the network hierarchy incurs higher delays than at the edge, evaluation results show that the increase in latency does not negate the performance gains of proactive caching. Moreover, the gains are amplified by the reduction in server load and cache redundancy is achieved.

Some research works address the mobility-aware proactive caching in IoV, [16,17,18,19,20]. In the paper [16], authors consider vehicular caching where vehicles are also caching devices and propose a vehicular caching scheme in content-centric networks, developing an optimization model which minimizes the network’s energy consumption. It uses the Lyapunov optimization method to minimize the online caching algorithm. Our work differs from that work because we focus on the minimization of network delay and we also address the issue of partial caching. Authors in [17] propose a solution for efficient utilization of caching resources of IoT devices and edge servers. It assesses the performance of the scheme using cache hit probability. However, that work does not consider the mobility of IoT devices, nor the issue of partial caching when an IoT node gets some part of the content from the currently connected Edge devices and the remaining part from a forthcoming Edge node.

The work in [20] formulates a joint optimization scheme for content caching and resource allocation for high-speed IoV and mobile edge computing. It considers the bandwidth of vehicle to infrastructure (V2I) and V2V links and the cache storage limit of edge storage and proposes a strategy to minimize the weighted average delay. It does not consider partial caching for high-speed vehicles but allows getting content from the forthcoming RSU. The work in [18] takes into consideration the mobility of users in a cache decision in the context of Cloud-based Radio Access Networks (C-RAN) and proposes a mobility-aware proactive caching strategy. It formulates cache placement optimization problems and minimizes the transmission delay. However, this work does not consider vehicular caching and the issues of partial caching for high mobility users.

Authors in [19] propose a hierarchical proactive caching scheme that considers vehicle and RSU caching. They formulate the optimization problem to minimize the vehicle communication’s latency. A very close related work [21] proposes a content caching decision optimization method in IoV to minimize the content fetching delay for vehicles considering the limited radio range and cache size of both caching vehicles and RSUs. It proposes an algorithm for vehicle association, content caching, and pre-caching decisions. The main contributions and comparisons to the most relevant and closely related published papers are shown in Table 1.

In summary, the key distinction among the related works in mobility-aware proactive caching is the assumption regarding exploiting vehicular caching to improve the cache hit and cache utilization by adapting ICN and V2V communication in the vehicular caching schemes. In the next section we present a mobility-aware proactive edge caching scheme, which to the best of our knowledge, has not been studied in the literature.

## 3. System Model

This section presents the proposed system architecture and concepts including network model, communication model, vehicle position and mobility model, caching, and caching decision model. Finally, the transmission latency optimization problem is formulated and solved using the dynamic programming method.

### 3.1. Network Model

In this study we consider an urban scenario that encompasses IoV, RSUs equipped with a wireless interface and storage, Cellular Macro Base Stations (MBSs), and a remote cloud center as shown in Figure 2 in the context of ICN networks. We assume that vehicles are communicating with RSU through V2I links and also with other vehicles through V2V.

The features of the ICN architecture make it a promising solution to fit the peculiarities and requirements of the IoV environment. One of the most important and helpful features of ICN is in-network caching, which allows users to acquire data from nearby servers to reduce latency. In this context, we leverage the Interest and data packets of ICN. The Consumer Vehicle (CV) sends an interest packet to get content, which might be cached on connected RSUs or MBS within its radio range, or otherwise located on a remote server. Edge Nodes that receive the interest packet and have the requested content in its caching storage deliver the data packets (content) to the CV. RSUs have the capability to serve multiple vehicles using their limited storage cache space. Each RSU and routing vehicle (RV) should exploit its limited resources (memory, CPU, cache space, …) proactively to provide optimal services.

IoV nodes in set $V=\{v_{1},v_{2},v_{3},\cdots ,v_{N}\}$ include both consumer vehicles (CVs), $CV=\{cv_{1},cv_{2},\ldots ,cv_{n}\}$ and router caching Vehicles (RVs), $RV=rv_{1},rv_{2},\ldots ,rv_{n}$ which act as caching node, RSU Nodes in set $R=\{r_{1},r_{2},r_{3},\ldots ,r_{n}\}$ placed along the roadside and connected to an MBS. We consider that each RSU *R* is equipped with an edge-cache, denoted as $R_{c}$, with the storage capacity of $R_{C}$ bytes, while the MBS is equipped with a cache, denoted as $M_{C}$ with a storage capacity of $M_{c}$ bytes. Important abbreviations and symbols used in this paper are listed in Table 2.

### 3.2. Vehicle Position and Mobility Model

Vehicles locations change over time while RSUs are assumed to be at fixed locations. Therefore, the mobility of the vehicle affects the transmission performance both in V2V and V2I links as the communication distance and trajectory changes over time.

#### 3.2.1. Modeling the Position

The position of the vehicles and the RSU at a given time is shown in Figure 3. Let *r* denote the radius of the RSU’s coverage, $h_{r}$ the height of the RSU antenna, $h_{v}$ the height of the vehicle’s antenna, $d_{rr}$ the distance from the RSU to the road, *T* the total time during which the vehicle is inside the coverage area of the RSU, (−T/2 < *t* < T/2) *t* the time counted from the point at which the vehicle is closest to the the RSU, *s* the vehicle’s speed (assumed constant), and *s.t* the distance of the vehicle from its position at t=0 to its current position. The vehicle enters the coverage area of the RSU at time *t* = −T/2, when its horizontal distance to the RSU is *r*. At *t* = 0, the vehicle is at a horizontal distance $d_{rr}$ from the RSU having moved a distance s.T/2. As time increases, the vehicle’s distance to the RSU increases until at time *t* = T/2 the vehicle is again at a horizontal distance $d_{rr}$ from the RSU, after which point it loses connectivity to the current RSU. Note the symmetry of the relationship of speed X time, derived by our choice of t=0. Considering additionally the effect of the difference between the heights of the RSU and vehicle antennas, we can see that the distance between the vehicle and the RSU $d_{vr}$ is given by Equation (Equation 1):(1)$$d_{vr}=\sqrt{d_{rr}^{2}+{\left(h_{r}-h_{v}\right)}^{2}+{\left(s.t\right)}^{2}}$$

It can be seen that at t=−T/2, the vehicle is just entering the coverage area and is at a distance *r* from the RSU, while at t=T/2 the vehicle is leaving the coverage area, situated again at a distance *r*, for |t|>T/2 the vehicle is outside the coverage of the RSU.

To model the relative position between the vehicles, we consider vehicles traveling in the same direction. V2V communication between vehicles traveling in the opposite directions is possible, however it is not sustainable for prolonged times, so the content transfer is limited. Let $d_{vv}$ be the distance between the consumer vehicle $V_{c}$ and the routing vehicle $V_{r}$ at time *t*, its value is given by Equation (Equation 2).
(2)$$d_{vv}=\sqrt{{\left[\left(O_{vc}-O_{vr}\right)+\left(s_{c}-s_{r}\right)t\right]}^{2}}$$
where $O_{vc}$ and $O_{vr}$ denote the coordinates of the starting positions and $s_{c}$ and $s_{r}$ denote the speed of the consumer vehicle and router vehicle, respectively. We assume that the vehicles speeds *s* are independent and identically distributed, generated by the following truncated Gaussian distribution (using Equation (Equation 3)) with mean μ, variance $\sigma^{2}$ with $s\in [s_{min},s_{max}]$.
(3)$$s=\left\{\begin{array}{cc} \frac{2exp(\frac{-{\left(u-\mu \right)}^{2}}{2\sigma^{2}})}{\sqrt{2\pi \sigma^{2}}\left(\frac{-{\left(u-\mu \right)}^{2}}{2\sigma^{2}}\right)-} & \mathrm{if}\;\left[s_{min}\leq s\leq s_{max}\right]\\ 0 & \mathrm{otherwise} \end{array}\right.$$

#### 3.2.2. Mobility Model

Understanding the mobility behaviour of the vehicles is essential for better resource allocation and management. In this work we model the mobility of the vehicles using a Markov model [22] in which the mobility behavior of the vehicle is predicted based on the observation of it over some period of time. Thus the mobility of the vehicles is a discrete stochastic process in which the probability of moving to a state (i.e., RSU) depends only on the previous visited state and the probability distribution of the transition between states. More specifically, it is composed of:A set of states $r=r_{1},\ldots ,rk$, in which each state corresponding to a frequent RSU is ranked in decreasing order;A set of transitions, which represents the probability of moving from state ri to rj. A transition from one state to itself can occur if the vehicle does not move during a certain time window.

The Markov model can be represented either as a graph or as a transition matrix. The mobility of the vehicle and its connection with RV or RSU can be modeled using a Markov process [22] as shown in Figure 4.

We developed a 2D Markov process to model the interaction of the consumer vehicle with the routing vehicle and RSU in the IoV network, as shown in Figure 5. Let *J* denote the set of consumer vehicle interest in the request queue, *K* the network connection condition between V2V and V2I where K=1, denotes the V2V connection and K=0, denotes the V2I connection. The consumer vehicle initially connects with the RSU.

Let p(k,j) denote the probability that request *j* is served by the network connection k,k=0 for connection with V2I and k=1 for connection with V2V. The network states in Figure 5 can shift from K=1 to K=0 and vice versa depending on the current network performance.

### 3.3. Caching Placement Model and Caching Constraints

We assume a set of contents $C=c_{1},c_{2},\cdots ,c_{n}$ served by IoV applications. Each content has a different size. It is assumed that each content in the system can be divided into $L_{c}$ chunks. The $L_{c}$ chunks can be pre-cached to the edge and/or on vehicles. We assume that the popularity of the contents follow the Zipf distribution [23], commonly used in similar works [24,25,26]. The probability that a vehicular user requests the content *c* from the set of contents *C* can be calculated using Equation (Equation 4).
(4)$$p_{c}=\frac{c^{-\gamma }}{\sum_{m=1}^{C}c^{-\gamma }}$$
where γ is the Zipf exponent, which characterizes the skewness of the Zipf distribution, thus caching the contents in the RSU that have a high probability of being requested by vehicles.

Due to the high mobility of vehicles, some consumer vehicles may not obtain the full content from the currently connected RSU and might require that the remaining amount be proactively cached in the next RSU. In such a case, the CV obtains part of the content chunks from the current RSU and will get the remaining from the next RSU. Thus, in order to reduce caching redundancy, it is not necessary to cache all chunks of the content in current RSU. Rather, it should cache only the parts of the content that the vehicle can get while it is within the coverage of the current RSU. Let $S_{f}$ denote the size of the content $c_{n}$ which can be divided into a number of chunks of size $S_{c}$. Let $L_{c}=\frac{S_{f}}{S_{c}}$ denote the number of content chunks, and $L_{min}$ the minimum number of content chunks that can be cached by the vehicle in range of the current RSU. If the content requested by CV is not cached by the RSU, the total data is given by Equation (Equation 5)
(5)$$S_{n,c}=L_{n,c}^{min}\frac{S_{f}}{L_{c}}$$

When the content requested by $CV_{n}$ is cached by the the RSU, the number of content chunks of *c* obtained through the V2I link within the coverage of RSU is denoted as $L_{n,c}^{CV}$ , and the data size is given by Equation (Equation 6)
(6)$$S_{n,c}^{CV}=L_{n,c}^{CV}\frac{S_{c}}{L_{c}}$$

In order to reduce caching redundancy only those chunks that can be fetched by vehicles within the coverage of RSU should be cached. Thus, $L_{c}^{CV}$ is Equation (Equation 7)
(7)$$L_{c}^{CV}=\max\left\{L_{1,c},\cdots L_{n,c}\right\}$$

### 3.4. Communication Model

In this part we describe the two communication modes, V2I and V2V, that the consumer vehicle uses. The consumer vehicle chooses only one mode at a time to get the content. We assume that V2I and V2V links use different frequency bands, thus avoiding interference between them. The backhaul link has a constant capacity. All vehicular antennas are omni-directional at a height of 1.5 m. The antenna height of RSU is assumed 9 m [27]. Let the link between vehicle *i* and RSU *r* and the link between vehicle *v* and vehicle *v* have bandwidths $B_{v,r}$ and $B_{v,v}$. The transmission delay for the consumer vehicle to get content from the caching vehicle is $D_{v,v}$, and the transmission delay from RSU is $D_{v,r}$.

#### 3.4.1. V2V Communication Link

Using V2V communication link, a vehicle can transmit its cached contents to the requesting vehicle. However, V2V transmission links are unstable due to the variable transmission range and high mobility of vehicles [28]. When the distance between the consumer vehicle $CV_{v}$ from the router vehicle $RV_{r}$ exceeds the communication range, the link will break, causing a disconnection between the CV and RV. In order to avoid this situation, the CV will fetch the required content from the RV only if CV can fetch all the contents from RV during the connection of a V2V link.

Let $R_{v,v}\left(t\right)$ denote the communication rate between the vehicles, which can be calculated using Equation (Equation 8)
(8)$$R_{v,v}\left(t\right)=B_{v,v}\left(t\right)\log\left(1+\frac{p*g_{v,v}}{N_{0}B_{v,v}}\right)\left(t\right),\forall v\in V$$
where *P* is transmission power, *B* is the channel bandwidth, $g_{v,v}$ is the channel gain between CV and RV, and $N_{0}$ is the noise power density. The latency from the caching vehicle is denoted by $t_{v,v}$ and is determined by the content size and data rate
(9)$$t_{v,v}=\frac{C}{R_{v,v}}$$

Let a binary variable $x_{v,v}$ denote the connection status between the consumer vehicle and the caching vehicle and a binary variable $\alpha_{cv}$ denote the caching indicator for content *c* on CV.
(10)$$x_{cv,rv}=\left\{\begin{array}{cc} 1 & \mathrm{if}\;CV_{v}\;\mathrm{and}\;RV_{v}\;\mathrm{are}\;\mathrm{connected}\;V2V\\ 0 & \mathrm{otherwise} \end{array}\right.$$
(11)$$\alpha_{c,cv}=\left\{\begin{array}{cc} 1 & \mathrm{if}\;CV_{v}\;\mathrm{cached}\;\mathrm{the}\;\mathrm{content}\;\\ 0 & \mathrm{otherwise} \end{array}\right.$$

Then, we can calculate (using Equation (Equation 12)) the amount of delay it takes to get the content chunks from the caching vehicle through the V2V link.
(12)$$D_{v,v}\left(t\right)=x_{v,v}\alpha_{v,c}t_{v,v,c}$$
where $x_{v,v}$ is the connection status between the consumer vehicle and the caching vehicle as defined in Equation (Equation 10), $\alpha_{v,c}$ denotes the caching indicator of routing vehicle *v* as defined in Equation (Equation 11), and $t_{v,v,c}$ is the latency between the consumer vehicle and the caching vehicle.

#### 3.4.2. V2I Communication Link

When the RSU gets a request from a vehicle within its coverage range, it transmits the precached content to the requesting vehicles through the V2I link. Let $R_{v,r}\left(t\right)$ denote the communication rate between the vehicle *v* and the RSU. Then, we have
(13)$$R_{v,r}\left(t\right)=B_{v,r}\left(t\right)\log\left(1+\frac{p*g_{v,r}}{N_{0}B_{v,r}}\right)\left(t\right)$$
where *P* is the transmission power, *B* is the channel bandwidth, $g_{v,r}$ is the channel gain between CV and RSU, and $N_{0}$ is the noise power density. If the vehicle requests data from its current RSU, the latency $t_{v,r}$ is determined by the content size and data rate following Equation (Equation 14).
(14)$$t_{v,r}=\frac{C}{R_{v,r}}$$

Let a binary variable $y_{v,r}$ denote the connection status between the consumer vehicle and the RSU and Let a binary variable $\beta_{c,r}$ denote the caching indicator for content *c* on RSU *r*.
(15)$$y_{v,r}=\left\{\begin{array}{cc} 1 & \mathrm{if}\;CV_{m}\;\mathrm{and}\;RSU_{r}\;\mathrm{are}\;V2I\;\mathrm{connected}\;\\ 0 & \mathrm{otherwise} \end{array}\right.$$
(16)$$\beta_{c,r}=\left\{\begin{array}{cc} 1 & \mathrm{if}\;RSU_{r}\;\mathrm{cached}\;\mathrm{the}\;\mathrm{content}\;c\\ 0 & \mathrm{otherwise} \end{array}\right.$$

Then, the delay for the consumer vehicle to get the content chunks from the RSU using the V2I link is given by Equation (Equation 17)
(17)$$D_{v,r}\left(t\right)=y_{v,r}\left(T_{v,r,c}+\left(1-\beta_{r,c}\right)T_{r,s,c}\right)$$
where $y_{v,r}\in \left\{0,1\right\}$ is the connection status between the consumer vehicle and RSU as defined in Equation (Equation 15), βc,r denotes the caching indicator of RSU *r* as defined in Equation (Equation 16), $T_{r,s,c}$ represents the delay for RSU to obtain a content chunk from the server via the backhaul link.

### 3.5. Problem Formulation

Considering the vehicle mobility, constraints of the caching capability of vehicles and RSU, and the deadline requirement of the vehicular user, the latency minimization problem is formulated in dynamic programming form. The model is formulated to find a cache policy to minimize the latency. In a proactive network, each RSU will cache contents protectively from the source. In order to cache contents for moving vehicles in IoV networks, we consider the mobility probability of the vehicle v joining RSU and performing content caching in each RSU (partial or full caching) depending on the speed of the vehicle and transition probability between RSUs. Thus, it alleviates the burden on the backhaul and reduces the caching redundancy in RSU. Let $q_{r,v}$ denote the mobility probability of vehicle *v* joining RSU and then the probability, $P_{r,v}$, of obtaining content $x_{r,v,c}$ from RSU is
(18)$$P_{r,u}=\sum_{r\in R}\left(q_{r,v}x_{r,v,c}\right)$$

Similarly, caching vehicles can contribute by caching some contents from the source and saving the network resources. Thus, given mobility probability, $q_{v,v}$, of vehicle *v* joining the caching and router vehicle *v*, then the probability, $P_{v,v}$, of obtaining content $x_{v,v,c}$ from the current connected vehicle is
(19)$$P_{v,v}=\sum_{v\in V}\left(q_{v,v}x_{v,v,c}\right)$$

When the content is not available at RSU or the caching vehicle, the interest packet is forwarded to MBS or data source. The probability, $P_{m}$, of obtaining data from MBS, and the probability, $P_{s}$, of obtaining data from the server can be calculated as
(20)$$P_{M}=x_{m,v,c}$$
(21)$$P_{s}=1-P_{v,v}-P_{r,u}-P_{M}$$

Therefore, an optimization problem to minimize the average transmission latency *T* of the network is formulated (Equation (Equation 22)).
(22)$$\begin{array}{cc} \min T\; & \frac{\sum_{v\in V}\left(t_{0}P_{r,v}+t_{0}P_{v,v}+t_{B}P_{M}+t_{s}P_{s}\right)}{\left|V\right|} \end{array}$$

Let $D_{s}$ denote the average latency incurred transferring content from the origin server to the MBS, $D_{m}$ is the average latency incurred transferring content from MBS to RSU, and $D_{v,c}^{1}$ is the total content acquistion delay of all consumer vehicles for the minimum content chunks when content *c* is cached by the RSU.

The content getting delayed to all consumer vehicles CV for the content chunks, when the content *c* is cached and not cached by RSU, is denoted as $D_{1}$ and $D_{2}$
(23)$$D_{1}=D_{v,c}^{1}+\sum_{i=1}^{I}D_{i,m,c}+\left(L_{i,c}-L_{i,c}^{*}\right)D_{s}$$
(24)$$D_{2}=D_{2}+\sum_{i=1}^{I}D_{i,m,c}$$

$D_{3}$ and D4 are the delays to get the content chunks cached in the next RSU for high mobility vehicles
(25)$$D_{3}=\sum_{i=1}^{I}D^{*}+\left(L_{i,c}-L_{i,c}^{*}\right)D_{s}$$
(26)$$D_{4}=\sum_{i=1}^{I}D_{i,m,c}^{*}$$

Finally, the objective function can be translated into the following functions:(27)$$\begin{array}{cc} \max\tilde{T}\; & \sum_{v\in V}\left(\sum_{r\in R}x_{v,r}\left(D_{1}-D_{2}\right)q_{ru}\right)+\sum_{cinC}x_{v,b}\left(D_{3}-D_{4}\right))\\ s.t.\;\; & C1:\sum_{v\in V}x_{r,v}Q_{v}\leq \min\left(Cr,Nr\right)\\ \; & C2:\sum_{v\in V}x_{b,v}Q_{v}\leq \min\left(CB,NB\right)\\ \; & C3:\sum_{r\in R}q_{ru}*x_{ru}+x_{bu}\leq 1\\ \; & C4:x_{v,r}\in 0,1,x_{v,b}\in 0,1 \end{array}$$
where $C_{r}$ and $C_{b}$ are the current cached sizes in the RSU and the Base station, *Q* is the requested data size in one time slot, $N_{r}$ denotes the data size that can be obtained from RSU during one time slot, $N_{b}$ represents the data size that can be obtained from MBS during one time slot.

The constraints (C1 and C2) in Equation (Equation 27) ensure that the content caching on RSU and Base station does not exceed the cache/storage capacity, C3 defines for all vehicular users that the total (sum) of cached data should not exceed the total to be cached. C4 establishes the caching indicator constraint.

The objective function and all the constraints are linear and decision variables are binary. Therefore, the optimal problem solution can be implemented in the available solvers. We can see that the problem can be solved using dynamic optimization, as its variables change from state to state or over time *t*. The optimization problem is the Multiple knapsack problem, which can be solved by the dynamic programming method. In the next section, we describe the proposed Algorithm 1.
**Algorithm 1:** Mobility-Aware Content Caching Algorithm
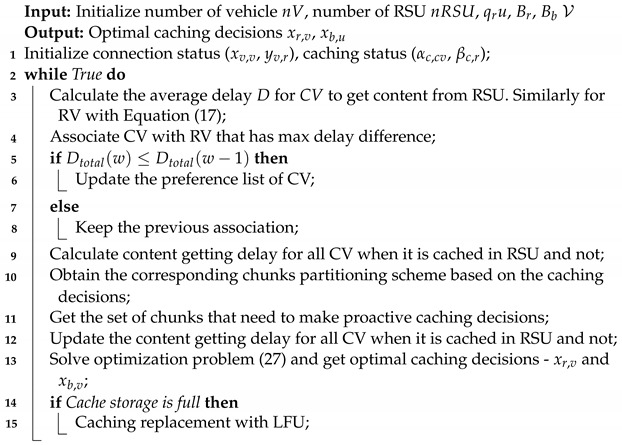


## 4. The Proposed Caching Algorithm

In this section, we propose a mobility-aware proactive content caching algorithm to solve the optimization problem formulated in the previous section and obtain optimal caching decisions. First, we formulate caching into a dynamic programming optimization model towards the minimization of communication delay. Then, we propose a proactive caching algorithm designed to meet the requirement of the IoV networks vehicle requirement.

### Mobility-Aware Proactive Content Caching Algorithm

Due to their mobility, vehicles may not get the complete content from a single RSU. In such a scenario, we need a proactive caching mechanism to cache some parts of the content that the current RSU is unable to serve and allow the remaining part to be cached in the forthcoming RSU. A mobility-aware proactive content caching scheme, to minimize the delay and improve caching hit ratio, is implemented in the following Algorithm 1. The operation of the algorithm has two parts: vehicles association (V2V or V2I) and optimal proactive caching. Initially, all consumer vehicles that need content *c* connect through the V2I link interface. Once the vehicle association is established, we determine the content caching delay of all CV for content chunks (minimum content chunks) to calculate the optimal caching decision. We used LFU cache replacement to clear space when the space in the Routing Vehicle or RSU is depleted.

## 5. Experiment Evaluation and Results

In this section, we set the main parameters, present the results of the optimization algorithm, and evaluate the performance of the proposed algorithm. The CacheAll, CachingMax, Random Caching, and Greedy Caching schemes widely used in the related works are introduced for comparison.

Random Caching: In this caching strategy, content is cached at any one of the RSU or caching vehicle nodes, and the node is selected randomly regardless of their popularity [29];Greedy Caching: In this caching strategy, the content is cached at the RSU or caching vehicle node, and the node is selected heuristically taking the locally optimal choice, disregarding any advantages of exploring and/or information gathering [30];CacheMax: The highest probability RSU caches contents [14];CacheAll: All adjacent RSUs cache contents.

### 5.1. Simulation and Parameter Settings

Considering a realistic urban IoV scenario with the IoV network having different number of vehicles ranging from low density (10 cars) to high traffic density (100 cars), 5 RSU, 1 Macro base station, and 1 cloud server in a 0.5 km × 0.5 km road segment. In the simulation scenario, for simplicity, we consider a two-lane multi-junction road segment as shown in Figure 2. For V2I communication, the coverage range of each RSU is 200 m and the vehicle can send Interest for contents once it is connected to one of the RSUs. We assume that a total of 5 RSUs are deployed at the 4-junction road, 1 RSU on each side of the junction, and 1 RSU at the center. Each RSU is assumed to cover the two-lane road and have some overlap radio coverage with a RSU located at the center of the road junction. Moreover, each RSU is connected to an MBS having a radio coverage range of 500 m.

We used Python and Pyomo [31], which is a Python based open-source optimization modeling language with a diverse set of optimization capabilities to build the simulation environment. We simulated the algorithm on a computer (Intel Core i7-7500U CPU @ 2.70 GHz × 4 and 12 GB memory) running Ubuntu 20.04 LTS operating systems.

The consumer vehicle node sends its request to its corresponding RSU, router vehicle, or cloud server. Vehicle nodes are associated with the edge node that has the smallest distance from them (i.e., has the smallest propagation delay). The communication between the IoV node and its corresponding RSU node is assumed to be by the means of the IEEE 802.11p protocol. Let the popularity of the content follow a power-law distribution. The consumer nodes send requests following a popularity distribution described by Zipf with α = 0.6. Every router vehicle has a cache and follows the LRU (Least Recently Used) replacement policy. We ran each experiment 100 times with different random seeds, then calculated the averages of the metrics. The main parameters used in the simulation are described in Table 3 and the source code is available from our GitHub repository (Simulation code: https://github.com/salubinseid/mobility-aware-caching-iov-icn, (accessed on 18 December 2021)).

### 5.2. Performance Metrics

We consider various performance evaluation metrics including Cache Hit Ratio (CHR), Latency, and Link Load to evaluate the performance of our scheme and compare with Random caching, Greedy caching, CacheMax, and CacheAll algorithms.

#### 5.2.1. Cache Hit Ratio

The cache hit rate metric is part of the request satisfied by the router (RSU or vehicles) nodes instead of the providers, thereby balancing the content requests among the available cache resources. The load on the provider (Server) will be substantially eased if users can acquire requested material from router nodes. As a result, the cache hit ratio is a crucial indicator for evaluating the ICN’s performance. The following Formula (Equation 28) is used to compute the cache hit rate:(28)$$CHR=\frac{\sum_{v=1}^{V}N_{v}}{R}$$
where *V* is the number of consumer vehicles (users) and $N_{v}$ is the cache hit counts of each consumer. *R* is the number of request from all consumers.

#### 5.2.2. Latency

Latency is the time between when the first interest packet for a content object is sent out and the corresponding data is received. In Equation (Equation 29), we calculate the average latency over *V* consumer vehicles that send *R* requests.
(29)$$Average\;Latency=\frac{\sum_{v=1}^{V}\frac{\sum_{r=1}^{R}D}{R}}{N}\;\left(\mathrm{ms}\right)$$
where *V* is the number of consumer vehicles (users) and *D* is the duration between when the first interest packet for a content object is sent out and when the corresponding data is received. *R* is the number of requests from all consumers.

#### 5.2.3. Link Load

Link load is another important performance metric to calculate the bandwidth utilization of the content of the network. It is the number of bytes of interest packet a link traversed per unit time to retrieve the requested content and can be calculated as [32]
(30)$$\begin{array}{cc} Link\;Load & =\frac{req_{size}*req_{linkCount}+count_{size}*content_{linkCount}}{duration}\;\\ duration & =content_{retrievalTime}-content_{requestTime} \end{array}$$
where $req_{size}$ is the size of interest packet in bytes, $req_{link_{c}ount}$ is the number of links traversed by the request to reach the source, $content_{size}$ is the size of the content to retrieve, and $content_{link_{c}ount}$ is the number of links traversed by the content to reach the requesting consumer vehicle.

### 5.3. Results

#### 5.3.1. Cache Hit Ratio

As an important performance parameter for ICN caching schemes, we used the Cache Hit Ratio to evaluate the performance of the proposed caching scheme under different parameters. A cache hit occurs when interest is fulfilled by one of the cache nodes (vehicular cache node or RSU) and a cache miss happens if interest can only be served by the original content source. Figure 6 shows the cache hit ratio with different caching sizes and with a different number of consumer vehicles. The result shows that the cache hit ratio increase as the cache size increases since more content can be cached by RSU caching vehicles. In addition, the cache hit ratio decreases with an increased number of consumer vehicles. Our caching scheme achieves the highest caching hit ratio compared to the other schemes in caching popular contents from the forthcoming RSU by predicting the mobility of the vehicle. Because each caching choice is made with a preset probability that cannot reflect the popularity of the contents, Greedy and CacheMax achieve a lower cache hit ratio than our technique. Due to its lack of awareness of the popularity of the contents and its aggressive caching method, CacheAll has the lowest cache hit ratio.

#### 5.3.2. Latency

Figure 7 shows the latency of our caching strategy and benchmark schemes showing the impact of the number of vehicles and cache size on the performance of expected delivery. One of the purposes of proactive caching is to improve the user experience by minimizing data retrieval latency. The proposed scheme performs better in comparison to the three caching strategies in all simulation experiment. As shown in Figure 7a, when the number of content requesting vehicles increase, the average delay of all caching schemes increases. It is obvious that as the number of consumer vehicle content demand increases, the traffic also increases, which in turn increases the latency. However, our scheme achieved the lowest delay compared with the other schemes. It is obvious that CacheAll caching scheme reduces latency as all the contents are cached on RSUs along the route of the vehicles. However, it has poor cache utilization and will fail with the increase in requesting vehicles and contents.

#### 5.3.3. Link Load Performance

We performed simulations for link load saturation rate caused by the tested algorithms. Thus, the saturation results can verify if the proposed algorithm reduces latency based on a higher network saturation. Figure 8 show the improvement in the link load performance of the proposed scheme.

#### 5.3.4. Impact of Vehicle Mobility

The speed of the vehicles determines the amount of contact time between the vehicles and its current RSU. High speed vehicles have a short contact time with their RSU.

Figure 9 shows the Cache hit ratio and Latency for various vehicle speeds. When the speed increases, which in turn decreases the contact time between the vehicles and RSUs, the vehicle cannot exploit the caching in the RSUs. Thus the content must be fetched from the server/base station, which will increase the expected time to receive the data.

## 6. Discussion

In this section, we discuss and interpret our findings from various perspectives. This study was conducted to design an optimal caching scheme in the context of ICN-based IoV networks, considering the mobility of vehicles. We evaluated the impact of various simulation parameters including cache size, speed of the vehicles, and the number of vehicles on the performance of our proposed scheme and related approaches. While the preliminary findings are encouraging under the proposed simulation scenario and assumption, there are still many issues, which are discussed as follows.

To provide proactive caching in a high mobility IoV environment, an accurate mobility model is required. Similarly to some previous works [22,33], we model the mobility of vehicles using a Markov chain, which requires collecting vehicular mobility information for accurate prediction. This generates a huge amount of data that is often collected centrally, raising privacy concerns that can constitute challenges for the practical implementation of mobility-aware caching. In order to address such issues, a distributed data collection and federated machine learning approach might be useful.

Furthermore, we assume that the popularity of the content is fixed for a certain period of time. However, the popularity of the content fluctuates quite often, which challenges the caching scheme. Thus, content popularity measurement mechanisms are required and therefore it is important to investigate how to implement adaptive caching, responding to the dynamic content popularity.

We also assume that each caching node (vehicles and RSU) has homogeneous caching and communication capabilities, however, under practical deployment, the catching capabilities of nodes might be different from one another and can change over time. It is thus important to investigate how to adaptively cache according to the dynamic nodes capacities, while also taking mobility into consideration.

## 7. Conclusions

In this paper, we propose a mobility-aware proactive edge vehicular caching scheme in ICN to improve IoV network performance. Mobility pattern and speed of the vehicles, content popularity and demand are vital for optimizing the proactive caching decision on RSUs. A 2D Markov model was used to model the communication between V2V and V2I. A cache decision problem is then formulated as a dynamic optimization model, aiming at minimizing the expected network delay and maximizing the caching hit ratio. Given the experimental results, we can conclude that our mobility-aware proactive edge caching scheme minimizes communication latency and maximizes the caching hit ratio compared with the benchmark caching scheme. Results show that caching in both RSU and vehicles has a significant performance enhancement when the number of vehicles and the caching size increases.

We considered a simulation scenario of a road network having a two-lane multi-junction road segment to study the effect of mobility-aware proactive caching on network performance. We plan to extend this work to consider a complete city model consisting of multiple roads, RSUs, and multiple MBS. Furthermore, we will focus on edge caching schemes under dynamic content popularity and joint optimization of computing resources of edge devices in addition to their caching storage to maximize the overall utilization of the systems of ICN-based IoV networks.

## Figures and Tables

**Figure 1 sensors-22-01387-f001:**
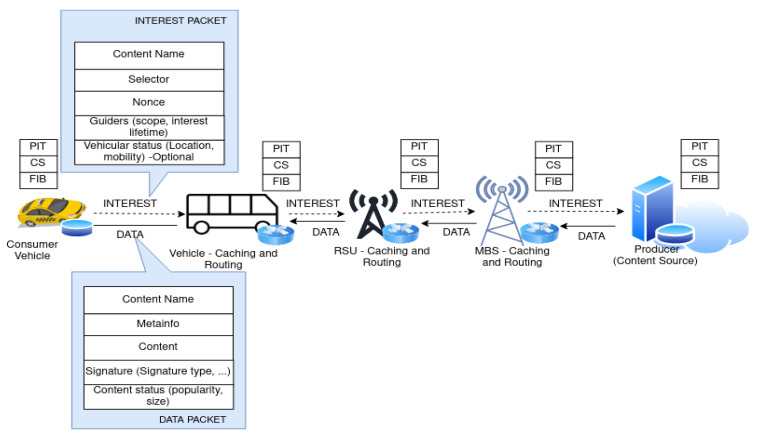
ICN in IoV.

**Figure 2 sensors-22-01387-f002:**
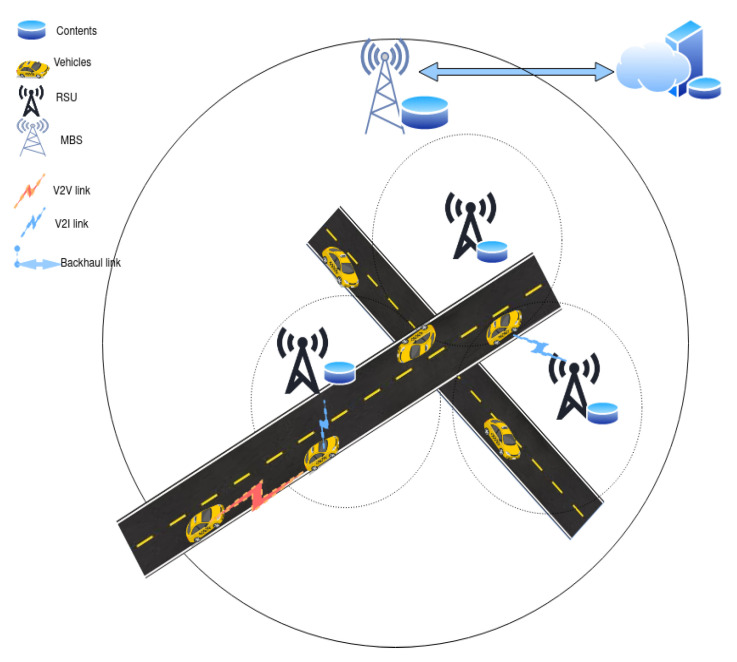
IoV networks scenario.

**Figure 3 sensors-22-01387-f003:**
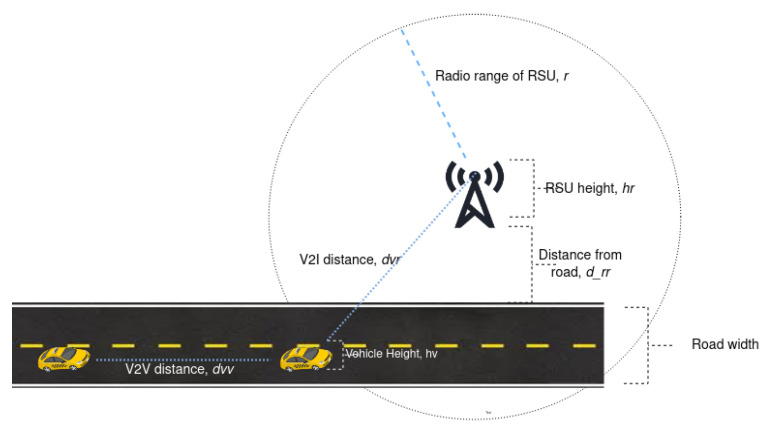
Position of vehicles and RSU.

**Figure 4 sensors-22-01387-f004:**
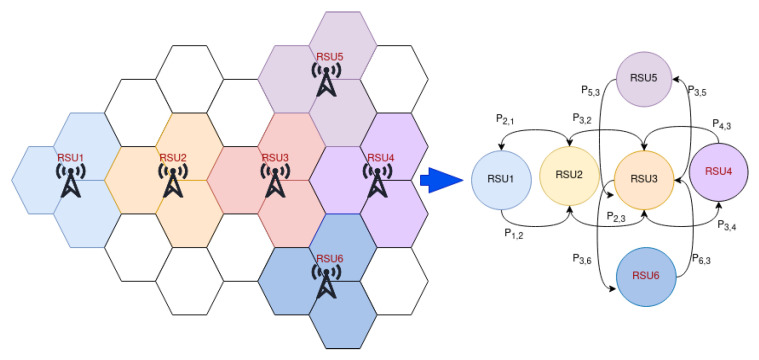
Roadside unit to Markov model.

**Figure 5 sensors-22-01387-f005:**
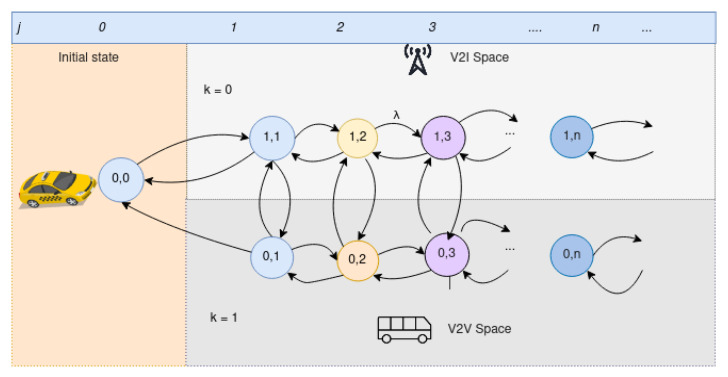
V2I and V2V 2D Markov model.

**Figure 6 sensors-22-01387-f006:**
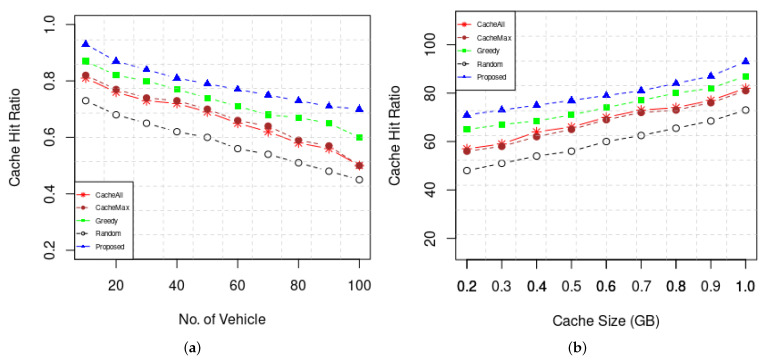
(**a**) Cache Hit ratio vs. No. of vehicles (**b**) Cache hit ratio vs. different cache size.

**Figure 7 sensors-22-01387-f007:**
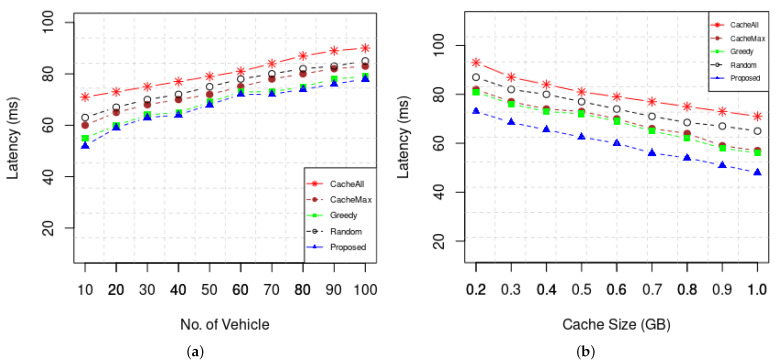
(**a**) Latency of caching vs. Number of vehicles (**b**) Latency vs. Cache Size.

**Figure 8 sensors-22-01387-f008:**
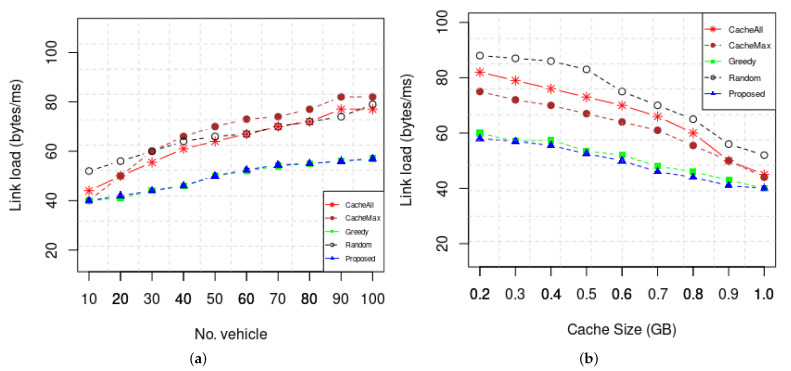
(**a**) Link load vs. number of vehicles and (**b**) cache size.

**Figure 9 sensors-22-01387-f009:**
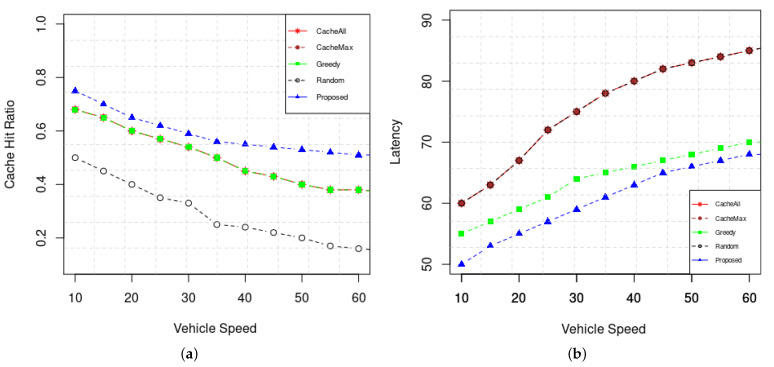
Analysis of Mobility impact (**a**) Cache hit ratio. (**b**) Latency.

**Table 1 sensors-22-01387-t001:** Related works.

Ref.	Network	Internet Archit.	Proactive	Communication	Caching At
[14]	Vehicular	ICN	Yes	V2I only	RSU only
[18]	C-RAN	-	Yes	User to RRH	RRH
[19]	VANET	TCP/IP	Yes	V2I only	RSU only
[21]	IoV	-	Yes Delay	V2V & V2RSU	RSU & Vehicle
This paper	IoV	ICN	Yes	V2V & V2RSU & V2BS	RSU, Vehicle & MBS

**Table 2 sensors-22-01387-t002:** Summary of key notations.

Notation	Definition
R	Set of RSU, $R=r_{1},r_{2},r_{3},\ldots ,r_{n}$
B	Set of Macro Base Station, $B=b_{1},b_{2},b_{3},\cdots ,b_{n}$
V	Set of vehicles, $V=v_{1},v_{2},\ldots ,v_{n}$
C	Set of contents, $C=c_{1},c_{2},\ldots ,c_{n}$
f	Chunks size
B	Bandwidth
r	Coverage radius of RSU
*s*	Speed of vehicle
$d_{vv}$	Distance between vehicles
$d_{vr}$	Distance between vehicle and infrastructure
γ	Zipf Exponent
t	Tolerance time

**Table 3 sensors-22-01387-t003:** Parameters.

Parameter	Value
No. of Vehicles	[10, 100]
Speed of Vehicles	[0, 60] km/h
No. of MBS	1
No. of RSU	5
Cache size of RSU	0.2 GB–1 GB
Cache size of RV *r*	0.2 GB–1 GB
Size of each content file $S_{f}$	40–100 MB
Size of each chunk $S_{c}$	2 MB
No. of Contents in library	[10, 100]
Popularity model γ	[0.6, 0.8]
Coverage of MBS	500 m
Coverage of RSU	200 m
MBS link capacity	100 Mbps
RSU link capacity	100 Mbps
Transmission power of vehicle	300 mW
Transmission power of RSU m/r	2 W
Noise power spectral density	−100 dBm/Hz
Request Pattern	Poission (λ=0.1)
Physical/MAC protocol V2I	IEEE 802.11p

## Data Availability

The data used to support the finding of this study are included within the article.

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
