# Peer review of "Mobility-Aware Proactive Edge Caching Optimization Scheme in Information-Centric IoV Networks"

_sensors, 2022, doi:10.3390/s22041387_

Round 1

Reviewer 1 Report

The article presents an Information-Centric Networks (ICN) based IoV networks design to provide delay-sensitive services on IoV networks. It is an interesting, relevant and pertinent article in the area of ​​mobility, mainly regarding IoV. The article has some aspects to improve, which are presented below:

  1. In the abstract at the end, present numerical values ​​of the difference obtained with the other works, changing the words "high" and "lower".
  2. Define the acronym, the first time the words appear and thereafter present only the acronym, do not rewrite the words of the acronym. Correct in all cases, two examples to correct are “RSU” and “ICN”, although it is recommended to review all the text.
  3. In the introduction section, I recommend changing "state of the art works" to "related works".
  4. In the introduction section, I recommend you summarize the last 2 contributions of the paper.
  5. In the first paragraph of section 2.1, one or a group of references appears to be missing.
  6. In section 2.2, I recommend you to better reference the works, not to mention "a paper".
  7. It is recommended that section 2.2 have a final paragraph or a table (with comparison parameters) where it is summarized what are the differences between the related works reviewed and the paper proposal.
  8. It is important that the Figures are referenced in the text, before presenting them in the document. For example, Figure 2 is presented before referencing it, this must be corrected. It is recommended to check that there are no similar cases.
  9. Regarding the writing of the text, please take into account what you (authors) did with verbs in the past tense in the text of the document. Differentiate what the "paper" presents, which you can mention in the present tense (but referring to the "paper" not you) and what you as researchers did, which you must mention in the past tense.
  10. To make your document look more organized, avoid literals A and B in different sections of the document. Use the numbering recommended in the MDPI model document.
  11. Check if the referencing of the equations is adequate, according to the MDPI model.
  12. In section 5 it is necessary to explain why you use the Random algorithms for comparison (Caching algorithm and Greedy Caching).
  13. It is also necessary to explain in this section why you selected the vehicle range from 10 to 100, 5 RSU, and 1 MBS.
  14. Check the text in search of typos, for example in the Conclusions section "We consider a realistic realistic urban IoV simulation".
  15. A "discussion" section is necessary, remember what the "instructions for authors" document mentions about it. "Authors should discuss the results and how they can be interpreted in perspective of previous studies and of the working hypotheses "
  16. The conclusion section should be improved considerably. You should not summarize what the other sections say.

Author Response

Thanks so much. We appreciate the time and effort to read our paper.

Reviewer 2 Report

The work presents a low scientific contribution. In addition, the authors' proposal is compared only with the Greedy and Random models. Therefore, I suggest that authors compare their solution with other models available in the literature. Along with this, I suggest that the authors do a more significant analysis of the current literature and thus provide a significant comparison of their work and other similar solutions. 

The first paragraph of the related works session seems to be disconnected from the rest of the related works session. Please consider removing the paragraph or rewriting it. 

The scenario used in the tests is specific and may not represent the characteristics of real scenarios. Furthermore, using a mathematical model to calculate vehicle routes reduces the scenario's ability to reflect a real environment. Therefore, I suggest that authors consider using a database with real vehicle traffic. Several databases provide an environment based on real traffic tracers on the Internet. 

Also, provide the network saturation rate caused by the tested algorithms. Thus, the saturation results can demonstrate if the proposed algorithm reduces latency based on a higher network saturation. It is worth mentioning that this network saturation must consider the content and interest packages. 

Please rewrite this sentence: "The dynamic nature of Internet of Vehicles and many other factors including device mobility, failure, network bandwidth, dynamic data exchanged demand the new Internet architecture and the current TCP/IP based networks cannot handle this growth." 

Please rewrite this sentence:" Test results show the methods successful in reducing total delay compared with the baseline in high mobility scenarios. "  

Please rewrite this sentence: "The cache amount adjusted by the edge cache. " 

Please rewrite this sentence:  “Our works are different from this work our focus on the minimization of network delay and also we consider the issue of partial caching.” 

Author Response

(The authors gave the same response as above.)

Reviewer 3 Report

It is a very well written and very well presented paper. It is written in a manner that will allow any person interested in the topic to understand it. In other words it is written with great care and precision. Congratulations!

Author Response

(The authors gave the same response as above.)

Round 2

Reviewer 1 Report

You have listened to all the suggestions and recommendations and have considerably improved the organization and content of the article.

Reviewer 2 Report

The authors have satisfactorily addressed most of my concerns.